# Association of Nighttime Sleep Duration with Depressive Symptoms and Its Interaction with Regular Physical Activity among Chinese Adolescent Girls

**DOI:** 10.3390/ijerph182111199

**Published:** 2021-10-25

**Authors:** Xueyin Wang, Jiangli Di, Gengli Zhao, Linhong Wang, Xiaosong Zhang

**Affiliations:** 1Department of Obstetrics and Gynecology, Peking University First Hospital, No. 1 Xi’anmen Street, Xicheng District, Beijing 100034, China; xueyin.wang@bjmu.edu.cn (X.W.); zhaogengli@sina.com (G.Z.); 2National Centre for Women and Children’s Health, Chinese Center for Disease Control and Prevention, Beijing 100081, China; dijiangli@chinawch.org.cn; 3National Center for Chronic and Non-Communicable Disease Control and Prevention, Chinese Center for Disease Control and Prevention, Beijing 100050, China; linhong@chinawch.org.cn

**Keywords:** depressive symptoms, sleep duration, physical activity, adolescents

## Abstract

Depression has become a major mental health concern among adolescents globally, and the relationship between depressive symptoms and nighttime sleep duration among adolescent girls remains unclear. This study aimed to examine the association between nighttime sleep duration and depressive symptoms among Chinese adolescent girls. This cross-sectional study, conducted in 2018, included 4952 girls aged 10–19 years from the eastern, central, and western regions of China. Depressive symptoms were assessed by the Patient Health Questionnaire-9, and categorized into depressive symptoms and non-depressive symptoms. Logistic regression models were used to estimate the odds ratios (ORs) of depressive symptoms. After adjustment for covariates, adolescent girls with a nighttime sleep duration of <7 h/night (OR = 2.28, 95% CI: 1.76–2.95) and 7 h/night (OR = 1.82, 95% CI: 1.48–2.24) were associated with increased risk of depressive symptoms, compared to those with a sleep duration of 8 h/night. An interaction between nighttime sleep duration and regular physical activity on the risk of depressive symptoms was observed (*p* for interaction = 0.036). Among both girls with and without regular physical activity, a sleep duration of <7 h/night was associated with increased odds of depressive symptoms, and the magnitude of the ORs among girls with regular physical activity was lower than those without regular physical activity. This study found a significant association of short nighttime sleep duration with increased risk of depressive symptoms, and demonstrates the importance of maintaining adequate nighttime sleep duration and ensuring regular physical activity in improving depressive symptoms among adolescent girls.

## 1. Introduction

Adolescence, a transitional period from childhood to adulthood, is characterized by physical growth and maturation of psychology and emotion; however, adolescents are also more likely to suffer from psychiatric illnesses including depression [1]. Depression has become a major public health concern across all age groups globally, and is one of the leading causes of disability and illness in adolescent populations worldwide [2]. In China, the overall prevalence of depressive symptoms in adolescents is 19.9%, ranging from 14.5% in the northeast to 23.7% in the central region [3]. Depressive symptoms in adolescents may contribute to cognitive and social impairments, increased risk of youth suicide, depression and anxiety in adulthood, and have adverse impact on quality of life, academic performance, and attainment of healthy autonomy and independence [4,5,6]. Moreover, gender differences in the rates of depression were reported to emerge between the ages of 13 and 15 years, and the prevalence of depression after puberty was nearly 2-fold greater in girls than in boys [7,8]. Therefore, research should focus more attention on the prevention and early identification of depression in adolescent girls.

Nighttime sleep duration plays a particularly vital role in maintaining physical and psychological health, and is closely correlated to various mental health problems in adolescents [9,10]. Both the National Sleep Foundation and the American Academy of Sleep Medicine have recommended a sleep duration of 8–10 h per night for teenagers [11,12]. However, a recent systematic review and meta-analysis focused on healthy children and adolescents aged 3–18 years revealed that pooled mean estimates for overnight sleep duration declined with increasing age, with adolescents aged 15–18 years getting only 7.4 h of sleep [13]. Previous studies investigating the relationship between nighttime sleep duration and depressive symptoms among adolescents remain inconsistent. Several epidemiological studies demonstrated that shorter sleep duration was associated with increased risk of depressive symptoms among adolescents [14,15,16,17], while there is also evidence suggesting no association between sleep duration and depressive symptoms [18]. Additionally, a cross-sectional study based on Japanese students reported a U-shaped association between sleep duration and depressive symptoms, showing that those who had a sleep duration that was too short or too long were more likely to have an elevated risk of depressive symptoms [19].

Physical activity has been proven to be an effective and low-cost intervention to prevent non-communicable diseases, improve mental health, and promote quality of life and well-being [20,21]. Physical activity not only has a beneficial impact on depression and anxiety, but it also contributes to a lower risk of cardiovascular mortality and improvement in the mental health of individuals affected with depression [22,23]. On the other hand, regular physical activity may also positively affect sleep by improving the endocrine system, the autonomic nervous system, and somatic functions [24]. Although several studies have examined the association between nighttime sleep duration and depressive symptoms, little is known about the interaction effect between nighttime sleep duration and regular physical activity on the risk of depressive symptoms in Chinese adolescent girls.

Therefore, the aim of the present study focused on evaluating the association of nighttime sleep duration and regular physical activity with the risk of depressive symptoms in Chinese adolescent girls, and further examining whether the association between nighttime sleep duration and depressive symptoms was impacted by regular physical activity. We hypothesized that shorter nighttime sleep duration was associated with increased risk of depressive symptoms, and this association was affected by regular physical activity.

## 2. Materials and Methods

### Study Design and Participants

The National Survey of Women’s Health is a cross-sectional, community-based study of women residing in three socioeconomic regions of China: the eastern (Jiangsu and Shandong provinces), central (Hunan and Anhui provinces), and western (Shanxi and Sichuan provinces) regions. Details of this study have been described elsewhere [25]. A multi-stage stratified random cluster sampling method was used to recruit participants. At the first stage, one district and one county were randomly selected as investigation sites in each provincial capital city of six provinces, and a total of 12 counties and districts were selected. At the second stage, one street community and one township were randomly selected in each selected district and county. A total of 24 street communities and townships were selected in this study. Lastly, in each selected street community and township, two communities and two villages were selected randomly and a total of 96 sampling areas were determined. In all these sampling areas, women aged 10–70 years were recruited. Face-to-face interviews were conducted to collect information on demographic characteristics, lifestyle factors, depressive symptoms, and school performance (only collected in girls aged 10–19 years) by using a structured questionnaire. Among the 5037 participants aged 10–19 years, the present analysis was limited to 4952 participants. Those with incomplete data collection for depressive symptoms (n = 58) or demographic/lifestyle characteristics (n = 28) were excluded. All participants provided written informed consent, and the study was approved by the Ethical Review Committee of the Chinese Center for Disease Control and Prevention, the ethic code was 201810.

## 3. Measurements

### 3.1. Assessment of Depressive Symptoms

The Patient Health Questionnaire-9 (PHQ-9), a brief, self-explanatory questionnaire, was utilized to evaluate the presence of depressive symptoms over the past 14 days [26]. The Chinese version of the PHQ-9 has adequate reliability and validity, and has been used by numerous studies focused on Chinese populations [27]. The PHQ-9 consists of nine items according to the diagnosis criteria of Diagnostic and Statistical Manual of Mental Disorders-IV, with a total score ranging from 0 to 27. The answer categories were based on a four-point response scale, with the categories ‘not at all’ (0), ‘several days’ (1), ‘more than half of the days’ (2), and ‘nearly every day’ (3). According to the sum of PHQ-9 scores, participants were classified into two groups: non-depressive symptoms (PHQ-9 score < 5) and depressive symptoms (PHQ-9 score ≥ 5) [28]. According to the sum of PHQ-9 scores, participants were classified into two groups: non-depressive symptoms (PHQ-9 score < 5) and depressive symptoms (PHQ-9 score ≥ 5) [29]. Participants with depressive symptoms were further divided into mild (PHQ-9 score: 5–9), moderate (PHQ-9 score: 10–14), moderately severe (PHQ-9 score: 15–19), or severe (PHQ-9 score ≥ 20) [29]. In this study, Cronbach’s α was 0.876, which suggested a strong degree of internal consistency.

### 3.2. Assessment of Nighttime Sleep Duration

Nighttime sleep duration was assessed by the question: “How long do you sleep on average per night?” The nighttime sleep duration was categorized into: <7, 7, 8 or ≥9 h/night, according to the 25th, 50th, and 75th percentiles of the nighttime sleep duration in this study.

### 3.3. Assessment of Regular Physical Activity

The frequency and duration of moderate-to-vigorous physical activity (e.g., brisk walking, jogging, cycling, and swimming) were obtained by asking “How often do you engage in moderate-to-vigorous physical activity on average every week?” and “How long do you engage in moderate-to-vigorous physical activity on average per time?” These items have been widely utilized to measure physical activity in adolescents [30]. Regular physical activity was defined as adolescents who engaged in moderate-to-vigorous physical activity at least 30 min per time, with no fewer than three times per week.

### 3.4. Assessment of Covariates

Covariates were assessed by using the interviewer-administered structured questionnaire including age (continuous), residence (urban, rural), education (primary school, junior high school, senior high school, college or higher), living arrangement (living with parents, living with grandparents/relatives, living in a dormitory, living with others), discretionary expenses (<100 RMB, 100–499 RMB, ≥500 RMB), drinking (yes, no), smoking (yes, no), interpersonal relationship (average or good, poor), and academic pressure (average or above, low). Smoking and drinking were defined as the participants who were currently smoking cigarettes or drinking alcohol.

## 4. Statistical Analysis

Demographic characteristics were presented as numbers and percentages for categorical variables, or median and interquartile range for skewed distribution continuous variables. The chi-square test and Mann–Whitney U test were conducted for categorical variables and skewed distribution continuous variables, respectively. Logistic regression models were conducted to estimate odds ratios (ORs) and their 95% confidence intervals (CIs) of nighttime sleep duration and depressive symptoms, as well as the interaction between nighttime sleep duration and regular physical activity groups. Models were adjusted for age, residence, education, living arrangement, discretionary expenses, drinking, smoking, interpersonal relationship, academic pressure, and investigation sites. The group with nighttime sleep duration of 8 h/night was used as the reference group. Further models included mutual adjustment of nighttime sleep duration or regular physical activity as appropriate. Furthermore, if the interaction items were significantly associated with depressive symptoms, stratified analyses were performed for adolescents who had regular physical activity and those who did not. Analyses were conducted using the SAS software version 9.4 (SAS Institute, Cary, NC, USA). All *p* values are two-sided, and a 0.05 level was used to declare significant differences.

## 5. Results

### 5.1. Characteristics of Study Participants

Among the 4952 adolescent girls aged 10–19 years, 20.7% (1027/4952) of participants reported experiencing depressive symptoms based on the total PHQ-9 score. The means of PHQ-9 scores were 0.93 and 8.52 for girls with depressive symptoms and those without depressive symptoms, respectively. Among the 1027 girls with depressive symptoms, 71.9% (738/1027), 18.2% (187/1027), 8.0% (82/1027), and 1.9% (20/1027) reported having mild, moderate, moderately severe, and severe depressive symptoms, respectively. Table 1 shows demographic characteristics of the study participants based on the presence of depressive symptoms. Compared to those who did not experience depressive symptoms, girls with depressive symptoms were older, more likely to reside in urban areas, have higher levels of education and discretionary expenses, and have shorter nighttime sleep duration (all *p* < 0.05). Adolescents with depressive symptoms were more likely to drink and smoke, and had a higher likelihood of having poor interpersonal relationship and elevated levels of academic pressure (all *p* < 0.05).

### 5.2. Association of Nighttime Sleep Duration and Regular Physical Activity with Depressive Symptoms

Figure 1 illustrates the rates of depressive symptoms by categories of nighttime sleep duration. Nighttime sleep duration was inversely associated with the risk of depressive symptoms (*p* < 0.05). Adolescent girls who reported less than 7 h/night of nighttime sleep duration had the highest rate of depressive symptoms (45.7%), while only 14.9% of those with nighttime sleep duration of ≥9 h/night experienced depressive symptoms.

Table 2 shows the association of nighttime sleep duration and regular physical activity with the risk of depressive symptoms. In the unadjusted model, adolescent girls with a nighttime sleep duration of <7 h/night (OR = 4.24, 95% CI: 3.38–5.31) and 7 h/night (OR = 2.16, 95% CI: 1.79–2.61) were associated with increased risk of depressive symptoms, compared to those with a nighttime sleep duration of 8 h/night. After adjusting for age, residence, education, living arrangement, discretionary expenses, drinking, smoking, interpersonal relationship, academic pressure, and investigation sites, the association of the nighttime sleep duration of <7 h/night and 7 h/night with depressive symptoms remained significant. After further adjustment for regular physical activity, adolescents who reported a nighttime sleep duration of <7 h/night were at the highest risk of depressive symptoms (OR = 2.28, 95% CI: 1.76–2.95), and those who reported a nighttime sleep duration of 7 h/night had an increased risk of developing depressive symptoms (OR = 1.82, 95% CI: 1.48–2.24). In contrast, regular physical activity was not associated with the risk of depressive symptoms in both the unadjusted and adjusted models.

### 5.3. Effect of the Interaction between Nighttime Sleep Duration and Regular Physical Activity on Depressive Symptoms

Table 3 presents the interaction between nighttime sleep duration and regular physical activity on the risk of depressive symptoms. There was a significant statistical interaction between nighttime sleep duration and regular physical activity on the risk of depressive symptoms in the unadjusted and adjusted models (unadjusted: *p* for interaction = 0.004; adjusted: *p* for interaction = 0.036; Table 2). We subsequently performed stratified analyses on participants who had regular physical activity and those who did not (Table 3 and Figure 2). In both groups, nighttime sleep duration of <7 h/night was statistically associated with higher risk of depressive symptoms after the adjustment for potential confounders, and the magnitude of ORs in adolescents with regular physical activity were lower than those without regular physical activity (adolescents with regular physical activity: OR = 1.81, 95% CI: 1.16–2.81; adolescents without regular physical activity: OR = 2.60, 95% CI: 1.88–3.58). Among adolescent girls without regular physical activity, the OR of nighttime sleep duration of 7 h/night for depressive symptoms was 2.22 (95% CI: 1.72–2.88). In comparison, the nighttime sleep duration of 7 h/night was not associated with depressive symptoms in adolescent girls reporting regular physical activity.

## 6. Discussion

In this large-scale, cross-sectional study of Chinese adolescent girls aged 10–19 years, 20.7% of adolescents reported experiencing depressive symptoms. The study found that shorter nighttime sleep duration (≤7 h/night) was associated with increased risk of depressive symptoms after the adjustment for potential confounders. In addition, the interaction between nighttime sleep duration and regular physical activity on the risk of depressive symptoms was observed. The magnitude of association between nighttime sleep duration of <7 h/night and depressive symptoms among adolescent girls with regular physical activity was lower than that of adolescent girls without regular physical activity. These findings generally confirmed the research hypotheses.

This study showed a high prevalence of depressive symptoms (20.7%) among Chinese adolescent girls aged 10–19 years based on the PHQ-9. Similar to this finding, a previous meta-analysis reported that the pooled prevalence of depressive symptoms was 19.9% among Chinese children and adolescents [3]. However, several previous epidemiologic studies focusing on different adolescent populations examined the prevalence of depressive symptoms with substantial disparities. An earlier cross-sectional study involving 2453 Japanese children and adolescents used the Birleson Depression Self-Rating Scale for Children (DSRS) to measure depressive symptoms, and reported a prevalence of 14.9% [31]. Moreover, the German National Health Interview and Examination Survey, based on data from 2863 German children and adolescents aged 7–17 years, found that 17% of participants experienced depressive symptoms assessed by the Center for Epidemiologic Studies Depression Scale (CES-DC) [32]. Recently, data from the 2017–2019 Korea Youth Risk Behavior Web-Based Survey of 87,355 South Korean adolescent girls indicated that 31.3% of girls suffered from depressive symptoms [33]. This disparity might be partly due to ethnicity factors, demographic differences (e.g., different age of study populations, inclusion of only adolescent girls in this study), discrepancy in the sample size, and the use of different measurements of depressive symptoms.

In line with previous research [34,35], this study also observed significant differences between adolescent girls with and without depressive symptoms regarding age, residence, education, socioeconomic status, smoking, drinking, interpersonal relationship, and academic pressure. These findings may contribute to early identification of adolescents who were susceptible to depressive symptoms. Furthermore, in order to explore a more accurate association of nighttime sleep duration and regular physical activity on the risk of depressive symptoms, the aforementioned factors should be considered as covariates.

In addition, we also found that shorter nighttime sleep duration (≤7 h/night) was associated with increased risk of depressive symptoms among adolescent girls. In concordance with our results, several previous studies also reported the significant association between short sleep duration and higher risk of depressive symptoms in both adolescents and adults. A prior longitudinal study based on 421 American adolescents suggested that adolescents with objective short sleep duration and insomnia symptoms were more likely to have withdrawn depressed complaints than those with normal sleep duration and with/without insomnia symptoms [36]. Recently, a cross-sectional study of 3724 Chinese adolescents revealed that sleep duration of <6 h/night was associated with a 2.39-fold increased risk of depressive symptoms compared to those with a sleep duration of 6–8 h/night [37]. Data from 1892 Korean adults selected from the 2014 Korean National Health and Nutrition Examination Survey showed that sleep duration of ≤5 h was correlated to increased depressive symptoms and a higher level of perceived stress [38]; another cross-sectional study of 1204 rural adults demonstrated that short sleep duration (<7 h per night) was associated with increased likelihood of depressive symptoms after adjusting for potential confounders. The potential mechanisms underlying the relationship between sleep duration and depressive symptoms may be partly interpreted as shorter sleep duration correlated to lower resting-state functional connectivity between the amygdala and regions involved in emotion regulation—including the ventral anterior cingulated cortex, precentral gyrus, and superior temporal gyrus [39]. This supported the theory that sleep-related alternations in corticolimbic circuity may contribute to the development of emotion-related problems including depression during adolescence [39].

One interesting finding of this study is that we found a significant interaction between nighttime sleep duration and regular physical activity in relation to the risk of depressive symptoms among adolescent girls. Specifically, compared to the group with a sleep duration of 8 h/night, the magnitude of ORs for the associations between shorter nighttime sleep duration and depressive symptoms in adolescent girls with regular physical activity was lower than those without regular physical activity. The results supported that physical activity might mitigate the adverse effect of short nighttime sleep duration on depressive symptoms among adolescent girls. Consistent with our findings, previous research also indicated the indirect impact of sleep duration on depressive symptoms through other influential factors. For example, Zhou et al. [37] demonstrated that the deleterious effect of short sleep duration on depressive symptoms may be altered by academic pressure. Overall, our findings highlight an important public health implication for advocating for regular physical activity among adolescent girls with short nighttime sleep duration in order to improve their depressive symptoms. In other words, regular physical activity among adolescents should be encouraged, and considered an effective intervention for enhancing the mental health of adolescents.

The main strength of this study was a relatively large sample size, which was helpful to more accurately estimate the magnitude of the association between nighttime sleep duration and depressive symptoms. In addition, the PHQ-9 was utilized to measure depressive symptoms, a validated instrument recommended by the Guidelines for Adolescent Depression in Primary Care to screen for depression in adolescents [40]. Moreover, potential confounding factors that could affect the relationship between nighttime sleep duration and depressive symptoms were adjusted. The study also had several limitations that should be noted. First, due to the cross-sectional study design, this study cannot determine the causality between nighttime sleep duration and depressive symptoms. Second, the study did not collect information on the use of anti-depressant medication, diet, bedtime, and wake time, so it was unable to evaluate the impact of these factors on the relationship between nighttime sleep duration and depressive symptoms among adolescents. Finally, we used self-reported nighttime sleep duration and physical activity rather than accelerometer-derived measurements, which may lead to recall bias.

## 7. Conclusions

In summary, this study found a significant association of short nighttime sleep duration (≤7 h/night) with increased risk of depressive symptoms after the adjustment for potential confounders. The interaction between nighttime sleep duration and regular physical activity on the risk of depressive symptoms was also observed. Moreover, the magnitude of association between short nighttime sleep duration (<7 h/night) and depressive symptoms among adolescent girls with regular physical activity was lower than that of adolescent girls without regular physical activity. Our findings emphasize the important public health implications for improving depressive symptoms, by advocating for adequate nighttime sleep duration and implementing interventions to encourage regular physical activity among adolescents. Further studies are needed to elaborate on the mechanisms underlying the interaction effect of regular physical activity on the relationship between nighttime sleep duration and depressive symptoms with more robust evidence, particularly among high-risk populations.

## Figures and Tables

**Figure 1 ijerph-18-11199-f001:**
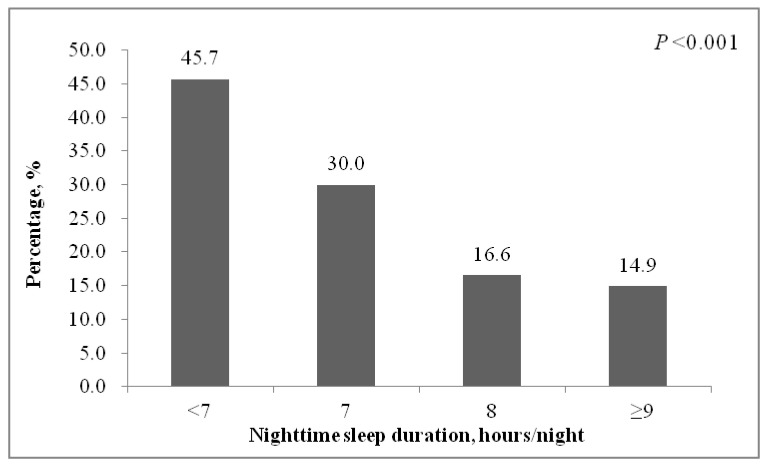
Percentage with depressive symptoms (the Patient Health Questionnaire-9 ≥5) by nighttime sleep duration.

**Figure 2 ijerph-18-11199-f002:**
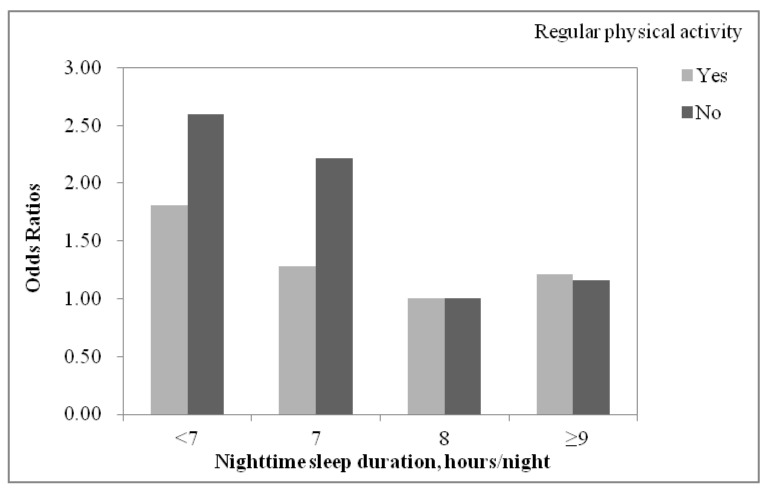
Adjusted odds ratios (ORs) for depressive symptoms by nighttime sleep duration, stratified by regular physical activity. Adjusted for age, residence, education, living arrangement, discretionary expenses, drinking, smoking, interpersonal relationship, academic pressure, and investigation sites.

**Table 1 ijerph-18-11199-t001:** Demographic characteristics of participants by the presence of depressive symptoms.

	Total	Non-Depressive Symptoms	Depressive Symptoms	*p* Value
Participants, n (%)	4952	3925 (79.3)	1027 (20.7)	
Age, years, median (IQR)	13 (11, 16)	13 (11, 16)	15 (13, 17)	<0.001
Residence				<0.001
Urban	2483 (50.1)	1889 (48.1)	594 (57.8)	
Rural	2469 (49.9)	2036 (51.9)	433 (42.2)	
Education				<0.001
Primary school or lower	1772 (35.8)	1564 (39.9)	208 (20.3)	
Junior high school	1541 (31.1)	1208 (30.8)	333 (32.4)	
Senior high school	1179 (23.8)	819 (20.9)	360 (35.0)	
College or higher	460 (9.3)	334 (8.5)	126 (12.3)	
Living arrangement				0.348
Living with parents	3797 (76.7)	2997 (76.4)	800 (77.9)	
Living with grandparents/relatives	442 (8.9)	350 (8.9)	92 (9.0)	
Living in a dormitory	626 (12.6)	512 (13.0)	114 (11.1)	
Living with others	87 (1.8)	66 (1.7)	21 (2.0)	
Discretionary expenses, RMB				<0.001
<100	2487 (50.2)	2033 (51.8)	454 (44.2)	
100–499	1316 (26.6)	1047 (26.7)	269 (26.2)	
≥500	1149 (23.2)	845 (21.5)	304 (29.6)	
Regular physical activity				0.413
No	3212 (64.9)	2557 (65.1)	655 (63.8)	
Yes	1740 (35.1)	1368 (34.9)	372 (36.2)	
Nighttime sleep duration, hours/night				<0.001
<7	416 (8.4)	226 (5.8)	190 (18.5)	
7	850 (17.2)	595 (15.2)	255 (24.8)	
8	2006 (40.5)	1674 (42.6)	332 (32.3)	
≥9	1680 (33.9)	1430 (36.4)	250 (24.4)	
Drinking				<0.001
No	4781 (96.6)	3842 (97.9)	939 (91.4)	
Yes	171 (3.4)	83 (2.1)	88 (8.6)	
Smoking				<0.001
No	4897 (98.9)	3896 (99.3)	1001 (97.5)	
Yes	55 (1.1)	29 (0.7)	26 (2.5)	
Interpersonal relationship				<0.001
Average or good	4601 (92.9)	3715 (94.6)	886 (86.3)	
Poor	351 (7.1)	210 (5.4)	141 (13.7)	
Academic pressure				<0.001
Average or above	3964 (80.1)	3297 (84.0)	667 (65.0)	
Low	988 (19.9)	628 (16.0)	360 (35.0)	

Values are median (IOR) or n (%).

**Table 2 ijerph-18-11199-t002:** Association of nighttime sleep duration, regular physical activity, and their interaction with depressive symptoms.

	Model 1		Model 2		Model 3	
	OR (95% CI)	*p* Value	OR (95% CI)	*p* Value	OR (95% CI)	*p* Value
Nighttime sleep duration, hours/night						
<7	4.24 (3.38, 5.31)	<0.001	2.28 (1.76, 2.95)	<0.001	2.28 (1.76, 2.95)	<0.001
7	2.16 (1.79, 2.61)	<0.001	1.82 (1.48, 2.24)	<0.001	1.82 (1.48, 2.24)	<0.001
8	Reference		Reference		Reference	
≥9	0.88 (0.74, 1.05)	0.166	1.18 (0.97, 1.44)	0.098	1.18 (0.97, 1.44)	0.098
Regular physical activity						
No	Reference		Reference		Reference	
Yes	1.06 (0.92, 1.23)	0.414	1.03 (0.88, 1.20)	0.755	1.03 (0.88, 1.20)	0.742
*p* for interaction *	0.004		0.036		N/A	

Values are odds ratios (95% confidence intervals). Model 1 was a univariate model. Model 2 was adjusted for age, residence, education, living arrangement, discretionary expenses, drinking, smoking, interpersonal relationship, academic pressure, and investigation sites. Model 3 was additionally adjusted for regular physical activity or nighttime sleep duration. * Nighttime sleep duration, regular physical activity, and the interaction item between nighttime sleep duration and regular physical activity were entered simultaneously into the multivariable logistic regression models. The *p* value for multiplicative interaction was presented.

**Table 3 ijerph-18-11199-t003:** Association between nighttime sleep duration and depressive symptoms stratified by regular physical activity.

Nighttime Sleep Duration, Hours/Night	Regular Physical Activity			
	No		Yes	
	**OR (95% CI)**	***p* Value**	**OR (95% CI)**	***p* Value**
<7	2.60 (1.88, 3.58)	<0.001	1.81 (1.16, 2.81)	0.008
7	2.22 (1.72, 2.88)	<0.001	1.28 (0.91, 1.81)	0.158
8	Reference		Reference	
≥9	1.16 (0.90, 1.49)	0.257	1.21 (0.88, 1.67)	0.241

Values are odds ratios (95% confidence intervals). Adjusted for age, residence, education, living arrangement, discretionary expenses, drinking, smoking, interpersonal relationship, academic pressure, and investigation sites.

## Data Availability

The data will be made available from the corresponding author upon reasonable request.

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
