# Peer review of "Association of Nighttime Sleep Duration with Depressive Symptoms and Its Interaction with Regular Physical Activity among Chinese Adolescent Girls"

_ijerph, 2021, doi:10.3390/ijerph182111199_

Round 1
Reviewer 1 Report
Dear Authors,
Your paper has impressed me a lot. Depression and sleep disorders have become a major public health concern across all age groups globally, including in adolescent populations worldwide, in particularly in girls. Wherein, nighttime sleep duration and physical activities are closely correlated to various mental health problems in adolescents including depression. There is a little known about the interaction effect between nighttime sleep duration and regular physical activity on the risk of depressive symptoms in Chinese adolescent girls that emerging and valuable for health care. I’m pleasure to say that I can recommend this important work for publication in journal “Journal of Environmental Research and Public Health”. As the same time, some comments should be corrected: 1) to structure the abstract to contain it Background, Methods, Results and Conclusions; 2) to revise the section “References” in accordance with the Instructions for authors; 3) it would be interesting to divide the study group (10-19 years old) by age, respectively with the periods of adolescence (early,10-13 years; middle, 14-17 years; and late, 18 and beyond), taking into account the different physiological need for sleep, different levels of physical activity and menstrual status for 10 and 19 year old adolescents. In this case, one could get some other interesting data. Please, if possible, present these results or make it the aim of your future research.
Good luck and further success.
Best regards, Reviewer
Reviewer 2 Report
The authors in this manuscript evaluate for associations with symptoms of depression in Chinese adolescent females, focusing on sleep duration and exercise. Using a large sample size, they find that lower sleep duration is significantly associated with symptoms of depression, and exercise reduced the risk of depressive symptoms in strata with lower sleep duration. These findings will be of interest to journal readers and the large sample size studied adds to the impact of the results. The results overall are easy to follow and understand. I recommend the authors address the following items prior to acceptance for publication:
- The label under Figure 1 does not clearly explain the results. I would suggest changing it to "Percentage with Depressive Symptoms (PHQ9 >= 5) by nighttime sleep duration."
- More information on the PHQ9 data would be helpful to interpret the results. Ideally, a breakdown of PHQ9 scores by different ranges of scores (number of subjects in the mild, moderate, moderate-severe, severe categories based on PHQ9 score), or at the very least, a mean PHQ9 score for the non-depressive vs. depressive symptom groups.
- Additional explanation is needed for why a PHQ9 cutoff of 5 was selected. While this is understandable to use this cutoff as scores below 5 are interpreted as essentially no symptoms of depression, the references cited use higher scores for a cutoff, including the paper cited as validating the the PHQ9, which uses a score of 10 as a cutoff. At the very least, additional rationale for why this score was chosen should be discussed.
Reviewer 3 Report
This study investigates the association of nighttime sleep duration with depressive symptoms and its interaction with regular physical activity among Chinese adolescent girls.
The topic of the manuscript is within the scope of the Journal and could be valuable to the scientific audience. The quality of the research design is acceptable.
Research hypotheses are not formulated.
TITLE
The title of the article is accurate.
ABSTRACT
Abstract reflects the work done and the conclusions drawn.
INTRODUCTION
Authors should formulate directional research hypothesis (-es).
METHOD
Some clarifications are however needed.
Please describe in details how multi-stage stratified random cluster sampling technique was used for sample selection.
Please justify, why regular physical activity was defined as adolescents who engaged in physical activity at least 30 minutes per time, with no less than 3 times per week?
RESULTS
The technique of data analyses seems appropriate.
DISCUSSION
Please discuss the confirmation or rejection of directional research hypothesis(-es).
TO SUM UP I think the author(s) need to make the recommended corrections.
Reviewer 4 Report
The manuscript is quite interesting and add some knowledge about depression, sleep, and physical activity. Here appears clearly the relationship between mental health and healthy habits, and how the behavioral changes may affect depressive symptoms in Chinese teenagers.
Nevertheless, I want to suggest to the authors that although it is a large sample, maybe a smaller number of participants, with more accurate determinations of physical activity (with accelerometers for example), or a more exhaustive anamnesis of each participant (medication, diet, bedtime and waking up time, etc.) could help to clarify some limitations of the study. A more exhaustive anamesis coul provide ideas about the origin of these depressive symptoms or even if poor sleep duration, low level of physical activity (and/or other variables) and depressive symptoms appear at the same time, or they are cause and effect.
Reviewer 5 Report
This is a straightforward study on the association of sleep duration with depression with, or without regular physical activity in adolescents. The authors found that a lower sleep time was associated with an increased risk of depression and that this risk was mitigated with regular physical activity. The introduction is clear and concise. This article was well written and easy to read. The conclusions were well justified based on that data. I only have minor suggestions for the improvement of the manuscript.
minor issues
- I suggest that figure 2 would be easier to understand and interpret as a paired bar graph with a legend as opposed to a 3 dimensional graph as it is.
- My apologies if the authors addressed this, but is there as association with regular physical activity and lower sleep times? It occurs to me that adolescents engaged in more organized sports, or physical activity might be too busy with academic and extracurricular activities that they do not have enough time to get adequate sleep. Is the conclusion supported that would suggest that adolescents should engage in regular physical activity at the potential cost of loss of sleep time to mitigate the potential for depressive symptoms?
